# Explainable Malware Detection System Using Transformers-Based Transfer Learning and Multi-Model Visual Representation

**DOI:** 10.3390/s22186766

**Published:** 2022-09-07

**Authors:** Farhan Ullah, Amjad Alsirhani, Mohammed Mujib Alshahrani, Abdullah Alomari, Hamad Naeem, Syed Aziz Shah

**Affiliations:** 1School of Software, Northwestern Polytechnical University, 127 West Youyi Road, Beilin District, Xi’an 710072, China; 2College of Computer and Information Sciences, Jouf University, Sakaka 72388, Aljouf, Saudi Arabia; 3Faculty of Computer Science, Dalhousie University, Halifax, NS B3H 4R2, Canada; 4College of Computing and Information Technology, University of Bisha, Bisha 61361, Saudi Arabia; 5Department of Computer Science, Albaha University, Albaha 65799, Saudi Arabia; 6School of Computer Science and Technology, Zhoukou Normal University, Zhoukou 466001, China; 7Faculty Research Centre for Intelligent Healthcare, Coventry University, Coventry CV1 5RW, UK

**Keywords:** malware analysis, transfer learning, malware visualization, explainable AI, cybersecurity, malicious, network behavior

## Abstract

Android has become the leading mobile ecosystem because of its accessibility and adaptability. It has also become the primary target of widespread malicious apps. This situation needs the immediate implementation of an effective malware detection system. In this study, an explainable malware detection system was proposed using transfer learning and malware visual features. For effective malware detection, our technique leverages both textual and visual features. First, a pre-trained model called the Bidirectional Encoder Representations from Transformers (BERT) model was designed to extract the trained textual features. Second, the malware-to-image conversion algorithm was proposed to transform the network byte streams into a visual representation. In addition, the FAST (Features from Accelerated Segment Test) extractor and BRIEF (Binary Robust Independent Elementary Features) descriptor were used to efficiently extract and mark important features. Third, the trained and texture features were combined and balanced using the Synthetic Minority Over-Sampling (SMOTE) method; then, the CNN network was used to mine the deep features. The balanced features were then input into the ensemble model for efficient malware classification and detection. The proposed method was analyzed extensively using two public datasets, CICMalDroid 2020 and CIC-InvesAndMal2019. To explain and validate the proposed methodology, an interpretable artificial intelligence (AI) experiment was conducted.

## 1. Introduction

The Internet of Things (IoT) platform was established to be the most user-friendly app in the industry. We can remotely access the actuators, seamlessly connect sensors, monitor smart devices in real time, and examine the information gathered via the cloud. While many of our existing devices rely on cloud computing, the IoT and app vendors are starting to investigate the benefits of using processing power from the devices themselves. This connectivity has one shortcoming: it is bidirectional. The cloud can make contact with a device that sends data there. Security over cloud sensors is a major concern because many IoT devices are made to be managed online. Chaos results if an IoT hacker has control over the devices [1,2]. With the growth of digital gadgets, we have entered the “mobile era,” with smartphones becoming increasingly popular. Powerful mobile technologies such as smartphones and tablets are replacing authoritative computational platforms, executing desktop computers. Apps that were previously only available on sophisticated computers are now available on these mobile devices. Android is the most popular smartphone operating system in the world, with over 80% of the market share [3]. Because of their popularity and ease of use, Android apps have seen a huge increase in the number of malware and security assaults directed against them. Android handsets are the most regularly attacked by malware, according to Nokia’s Threat Intelligence Report 2019 [4]. In the mobile service, the survey [5] discovered that Android was targeted by 47% of malware samples, while Windows was targeted by only 35.82%. The problem of identifying and combating malicious attacks must be given attention in light of the massive growth in the number of Android smartphones and Android malware.

Malicious apps employ a variety of techniques to circumvent the current detection systems supplied by the Android platform and existing anti-virus tools. Dynamic execution, code obfuscation, repackaging, and encryption are examples of evasion techniques [6]. These methods mostly work when dealing with APK meta-information (execution file, source code, permission, manifest files, etc.). However, these methods should be re-designed when dealing with network-based malware. A recent study [7] investigated a wide range of malicious activities and categorized current malware detection systems into two categories: static analysis and dynamic analysis. Static analysis is vulnerable to malicious code polymorphism and encryption [8,9], which are used to create versions of malware to avoid detection. Dynamic analysis techniques change the operating database of the device in real-time to control and access sensitive data. This idea is promising, but it requires a large enough set of actions to encompass app behavioral patterns. As a result, the dynamic analysis of smart devices with limited resources is challenging.

### Problem Statement

Several malware detection techniques [10] concentrate on the network traffic produced by Android apps. The goal of network traffic-based approaches is to uncover distinguishing features that can be used to classify malicious apps effectively. To harm a target Android app, network malware may use multiple malicious scripts. Text-based feature analysis can uncover potentially harmful scripts in terms of behavioral segmentation. Figure 1 shows malicious network behaviors using adware and SMS malware. Some malicious software connects directly to an IP address without resolving the address. This is also commonly regarded as a malicious indication. In part a, the IP with 10.42.0.1 is malicious as it works without Address Resolution Protocol (ARP). However, the IP with 10.42.0.151 used the domain resolution, which indicates that this is a valid address. In part b, the SMS malware continuously sends malicious scripts to the server for authorized access and payment. Such behaviors cannot be interpreted solely by image representation. However, the text-based analysis may encounter challenges such as code obfuscation, insertion, re-ordering, etc. Image-based malware identification is frequently utilized because it can capture various forms of structural data, including storage, processes, headers, etc. As a consequence, malware images can be used to extract any kind of dynamic or obfuscated data. However, this alters the entire design of network data packets, making it difficult to identify a particular script, including a malicious IP, script, URLs, etc. Additionally, this method is completely reliant on image properties. Therefore, an attacker can target the malware image, compromising the overall classification accuracy. To address these challenges, we integrated text-based features to detect potentially malicious scripts with image features to detect other potentially harmful behaviors such as storage or resource use. A hybrid strategy facilitates the appropriate classification and usage of malware and benign data [11].

In this paper, we proposed a novel and innovative approach to finding and evaluating network malware. We combined the trained textual and texture features to achieve the benefits of both. We demonstrated that these two types of features complement one another and that integrating them can improve malware detection results. The main contributions of the paper are as follows:The BERT-based transfer learning approach was utilized to extract trained features from HTTP and TCP flows. BERT employs transformers and an attention method that discovers contextual relationships between features and generates trained matrices.The malware-to-image transformation method was developed to convert the network byte streams into a visual representation. Further, the FAST extractor and the BRIEF descriptor were used to locate and mark essential features quickly and easily.The trained textual and texture features were combined for accurate Android malware classification.To explain and validate the proposed approach, an interpretable AI approach was used.

The remaining part of the paper includes the following: Section 2 describes the related work, and Section 3 describes the proposed method. Section 4 thoroughly discusses the experiments and discussions, and Section 5 concludes the work.

## 2. Literature Review

Numerous research studies [12,13] show how the Android operating system uses several security procedures, particularly authorization processes, to safeguard compromised target devices. To the advantage of admin rights, people must be sufficiently knowledgeable about security vulnerabilities. Android malware can infect and spread via mobile devices due to the limitations created by overreliance on the client. Internet connection is increasingly reliant on smartphones. Network management tasks are made easier by analyzing mobile app traffic. Several malware detection algorithms focus on the network traffic generated by Android apps. The purpose of network traffic-based techniques is to find distinctive characteristics that may be utilized to efficiently classify harmful apps. 

By monitoring the app’s uses-permission and utilizing metadata, Sanz et al. [14] created a static-based technique that effectively identified malware. The planned work had a classification performance of 86.41 %. Using the Drebin dataset, Puerta et al. [15] utilized a similar technique to identify security attacks and found a 96.05% classification performance. A two-phase malware analysis approach was presented by Liu et al. [16]. The first step is to examine the app’s Manifest.xml file, which contains the rights that have been sought. The second step is to use APK utilities to segment the file and retrieve the Smali code. Details regarding stated rights, particularly API calls, may be found in the Smali code, which can be utilized to identify suspicious behavior. The predictive accuracy of the proposed methodology is 98.6 %. For irregular network analysis, Shanshan et al. [10] suggested an HTTP and TCP-based malware detection method. The data flow of the handy app is replicated by the access point. All data acquisition and malware detection take place via the internet, using the possible available resources. Mobile malware is identified with a 97.89% accuracy using network-based traits and stochastic gradient descent. Aresu et al. [17] looked into HTTP-based datagrams generated by Android apps when they interface with remote harmful services. It also employs a grouping mechanism for creating profiles from many malware strains. These identifiers are then used to identify anomalous activities. 

Malware visualization is a well-studied topic that encompasses a wide range of methodologies in various applications. Nataraj et al. [18] initially visualized malware as gray-scale images in the range [0, 255], where 0 is black and 255 is white. They noticed that the images contained multiple parts that reflected different malicious content. The image descriptor was utilized to quantify malware texture characteristics and K-NN for classification. Wang et al. [19] created the TextDroid approach, which splits the text of an HTTP flow into special symbols and then generates n-gram sequencing to investigate the pattern of the resultant properties. The detection score for this text-based approach is 76.99%. Wang et al. [20] transformed network traffic into 2D images, and then a CNN network was utilized to classify the malware based on the visualized data. It did not derive attributes from traffic but used raw traffic patterns as images to conduct malware classification. The proposed technique has a classification accuracy of 99.9%. The Falcon [21] approach was proposed for malware classification based on network-to-image. Each network packet is treated as a 2D image for classifying network traffic. They employed a bidirectional LSTM network to process 2D images to obtain meaningful vectors for malware classification. The proposed method provides the malware classification of 97.16%.

A malware image reflects the malicious characteristics of each variant. However, image-based malware classification is limited to image attributes. As a result, a hacker can assault the malware image, altering the overall classification performance. Similarly, utilizing a text-based approach for malware classification alone may result in identifier renaming, re-ordering, and obfuscation issues. To address these concerns, we combined textual and visual features to design an efficient network-based malware classification and detection system.

## 3. Proposed Methodology

Android malware has grown to be a major issue in recent years. Current approaches can identify malicious apps reliably by observing the activities of mobile apps. However, mobile devices are often resource-constrained. By mining network traffic features, cloud services can be utilized to detect Android malware. Thus, it can help minimize the burden on mobile devices. The proposed method employs transfer learning, i.e., transformers, to extract the most meaningful features from network traffic. This reduces the training time required to train the model on large datasets. Furthermore, the development of a network-based malware detection system is less complicated. For instance, such a technique can be installed on a cloud server, relieving the unnecessary strain on mobile devices. These solutions are based solely on consumer data over cloud servers, ensuring access to mobile apps. The network communication traces enable the tracking and detection of various malware types [22]. Figure 2 describes the explainable malware detection system using transfer learning and texture features analysis. The Android network traffic was analyzed, and the encoded information was retrieved in the form of packet capture files. We analyzed data traffic in two directions: textual and visual feature analysis. The most prominent features were extracted from a large volume of hybrid features. In addition, the CNN model was intended to extract deep features. The parameters were fine-tuned to use the most efficient number of input and output layers, hidden layers, neurons, dropout layers, and activation and output functions. This procedure can assist us in lowering the computational cost of the CNN model and obtaining the most effective deep features for the ensemble model.

### 3.1. Textual Features Analysis

#### 3.1.1. Network Data Pre-Processing

HTTP traffic was utilized since it is the most popular protocol for global communication. The information contained in HTTP headers can be utilized to detect suspicious attacks. Mobile apps, on the other hand, communicate via encoded HTTP, making it impossible to obtain confidential content. To tackle this limitation, we used a combination of HTTP flows and TCP streams for collecting effective features from PCAPs. PCAP files are the primary records that are created during network data transmission. These documents contain network traffic used to evaluate the malicious node communication process. They also aid in network traffic planning and activity sensing. HTTP traces include source, destination, port, host, source info, bytes, packet length, frame length, and TTL. GET, POST, and URLs such as “www.google.com” are in the source info. TCP flows include transmitted and received data and overall session counts throughout conversations. Valuable information can be filtered to preserve semantics. To prevent redundant data, attributes from input sequences that were consecutively similar were eliminated. Short patterns were removed because they may not provide enough data to recognize network activity. Unifying sequence is crucial for detecting attacks because distinct pattern dimensions mislead neural network algorithms. To adjust the dimension, this method employs a predefined sequence length L. Sequences longer than L retain their first L names, whereas those shorter than L are unified by zero-padding.

#### 3.1.2. Transfer Learning with BERT

BERT is an Apache 2.0 licensed Natural Language Processing (NLP) machine learning framework. It is a pre-trained model used to decipher the meaning of long and complex texts. BERT employs transformers, a deep learning model in which each outcome element is connected to each input and the attention head between them is computed dynamically [23]. It analyzes the document as a whole instead of chronologically. The BERT bidirectional model is named after the fact that it can compute the right and left contexts of words in this way. Non-contextual models generate only a word description, irrespective of how the term is used in the document. For instance, the term “match” may have the same interpretation as the phrases “match the words” and “light the match.” BERT, a contextual model, generates various interpretations for terms that are connected to other terms in the phrase [24]. We employed a BERT-based model for word embedding and transfer learning from network traffic. It employs 12 layers of transformer blocks, has a hidden size of 768, 12 self-attention heads, and approximately 110M trainable parameters. The BERT-based mapping of the features is depicted in Figure 3. A sequence of embedded network features (w_1_, w_2_, etc.) are processed by the BERT-based neural network. Each of the resultant H-dimensional vectors corresponds to an input feature with the same index. Before feeding each feature sequence into BERT, 15% of the features were replaced with [MASK] tokens. The relevance of the non-masked features was used by the model to forecast the current value of the masked features. We used the following parameters for the BERT output feature prediction [25]:A classification layer was added on top of the encoder output.By multiplying the output vectors by the embedding matrix, the lexical dimension was made from the output vectors.The probability of each feature in the vocabulary was calculated with the help of the softmax method.

Figure 4 shows the visualization of features for the corresponding attention head using the BERT model. Part a shows the head view visualization with different layers. Part b shows the neuron view visualization of query q and key k. Individual neurons in the q and k vectors are visualized in the neuron view, which demonstrates how they are utilized to evaluate attention. Before entering the model, the input was processed to assist with it discriminating between the two network flows.

A [CLS] token begins the first network flow and a [SEP] token ends each network flow.Each feature has a network flow embedding signifying network flow A or network flow B. Network flow embeddings are conceptually similar to word embeddings with a vocab of 2.Each feature is given a positional embedding to denote its place in the sequence.

### 3.2. Visual Features Analysis

We examined a malware detection method based on visual features because malware is often updated to evade static and dynamic classification. In this method, the malware file is turned into an image, and the texture characteristics are extracted. It does not require malware signatures or reverse engineering. Anti-detection methods such as signature manipulation and dynamic feature extraction evasion can be effectively combated with this strategy [26]. The PCAP is explored to collect the byte stream about each malware variant. We designed a malware-to-image transformation method capable of retrieving images from byte streams. The 8-bit vectors from network-based byte streams are processed to produce gray-scale malware images. After that, the image sizes are all set to 25T56 pixels. Figure 5 shows a selection of 256 × 256 malware images for botnets, premium SMS, ransomware, and scareware. It was revealed that a huge PCAP size is reduced to a smaller image size. For instance, the PCAP is transformed from megabytes to kilobytes in the image. As a result, it may be possible to reduce computing power.

The texture features were then extracted from malware images using the combination of FAST extracted and BRIEF descriptor [27]. The FAST extractor has efficient real-time computation. First, it circles a pixel (p) with 16 pixels, termed the Bresenham circle, to detect corners. Here, we identified pixels from 1 to 16 and checked random N labels in the circle if the labeled pixel was brighter than the 16-pixel selection. BRIEF is only a feature descriptor; therefore, the FAST corner extractor was used for feature extraction, and BRIEF was utilized for feature description. The implementation process was divided into 3 steps for ease. To begin, the image was loaded into memory, and a copy was created that was unchanged in terms of scaling and rotation. Then, the FAST extractor and BRIEF descriptor were used to mark features. After, feature points between images were tracked, as shown in Figure 6.

### 3.3. Class Balancing Using SMOTE

A problem of class imbalance could arise because we are dealing with two distinct types of features, text and images. We encountered a problem of class imbalance in the combined features of textual and texture during the training. The combined dataset included a wide range of characteristics in text and images. When one class dominates another, it can be challenging to train the classifier for each class equally. It has a substantial impact on the evaluation criteria and classification accuracy. During training, the classifier may learn enough about the major class and ignore the lower, resulting in improved accuracy for major but poor accuracy for lower. We used the Synthetic Minority Over-Sampling (SMOTE) technique to oversample minority classes to fix class imbalance [28]. It incorporates new minority class samples based on their similarity to the original minority class samples. This causes the influence of minor classes to approach that of major classes. SMOTE works as follows:


It calculates the k-nearest neighbor value for each minority class $x_{i}\in S_{\min}$ using Euclidean distance.It chooses a random closest neighbor $x_{j}$ in a group of the k-nearest neighbor $x_{i}$. A new sample was produced based on Equation (1).


(1)$$x_{\mathrm{new}}={\;x}_{i}+\left|x_{i}+x_{j}\right|+\;\delta$$
where δ [0, 1] is a random factor that controls the placement of newly generated samples.

### 3.4. CNN-Based Deep Features Extraction

CNN mines a huge number of features to extract deep and significant features that reduce the classification model load and processing capability. To perform this, CNN supplied trained textual and visual texture information. CNN is employed in several malware studies [29,30]. CNN’s model works best with text, graphics, and video. We employed a one-dimensional CNN network with convolutional layers, pooling layers, dropout layers, and a fully connected layer. Convolution acts as a filter, cycling over combined features and producing the optimal feature interpretations. Each filter creates a feature map. Hyper-parameter changes are used to identify the ideal number of filters. Three convolution layers were utilized, each having 64, 128, and 256 filters. Max-pooling minimizes the size of the feature space, the range of features, and the computation complexity. This layer also creates a feature map using the most important features from the previous set. In addition, we used the batch normalization layer from the Keras framework with the CNN network. The resultant mean and standard deviation were kept close to 0 and 1 via batch normalization. It also reacted differently during training and validation. The learning process was thus stabilized, and the number of training epochs required by deep networks was reduced. Softmax and dropout layers in the proposed CNN network combatted overfitting. The output of the CNN network is represented by Equation (2).
(2)$$o_{k}^{1}=f(c_{k}^{1}+\sum_{i=1}^{N_{l-1}}\mathrm{Con}1D\left(X_{\mathrm{ik}}^{l-1},{\;t}_{i}^{l-1}\right))$$
where $c_{k}^{1}$ is the parameter bias of the kth neuron in the first layer, $t_{i}^{l-1}$ is the outcome of the ith neuron in layer l−1, $X_{\mathrm{ik}}^{l-1}$ is the kernel strength from the ith neuron in layer l−1 to the kth neurons in layer l, and “f()” is the activation function. After studying the detailed information, we selected the 400 most prevalent features for appropriate malware classification.

### 3.5. Voting-Based Ensemble Learning

For effective malware classification, the deep and prominent features were input into a soft voting-based ensemble model. The ensemble was a powerful model produced by methodically combining base techniques. In soft voting, each independent classifier provides a statistically significant indication that a given data point belongs to a particular class label. This enables more progressive and decentralized decision-making. The predictions were summed after being weighted by the importance of the classification model. The vote was then given to the target class label with the highest sum of normalized probabilities. The ensemble model can handle classification and regression challenges that individual models are unable to [31]. The suggested study employed a soft polling ensemble method. To begin, we used training data to construct the basic models of the Gaussian Naive Bayes (GNB), Support Vector Machine (SVM), Decision Tree (DT), Logistic Regression (LR), and Random Forest (RF) classification approaches. After that, the efficacy of these models was confirmed by employing test data, with each model delivering a unique assessment. Ensemble learning utilizes the estimations of several different procedures as supplemental information to achieve the desired level of final classification performance [32]. The complete process of the proposed method is given in Algorithm 1. It describes the overall procedure for the proposed study. The network traffic was provided as input in the form of PCAPs, and the malware classification was delivered as output. The PCAP file was filtered for the required TCP and HTTP information. The BERT-base model was intended to extract train features from the combination of TCP and HTTP. PCAP bytes were mined and converted to images to extract texture features using FAST and BRIEF. Textural and texture features were combined and fed into the soft-based voting ensemble model for effective malware detection and classification.
**Algorithm 1: Malware Classification Using Transfer Learning and Texture Features**InputPCAPOutputMalware ClassificationStep 1:Set P= {$p_{1}$, $p_{2}$, …, $p_{n}$}s, where is P is a PacketsStep 2:$\mathrm{Decrypt}\;\left(P\right)=\;P\prime$
Compute PCAP from  P′, where  P′ $=\left(\mathrm{IP},\;\mathrm{TCP},\mathrm{HTTP},\ldots ,n\right)$
Selects NF from PCAP, where NF is the required Network Flows
Display/Select HTTP + TCP Step 3:Select HTTP traces and TCP flowsStep 4:Tokenize and filter HTTP and TCP flows = Clean features Step 5:Apply word embedding
BERT transformers = Train feature
 Extraction = Textual trained featuresStep 6:Trained files = Trained features as TStep 7:Compute B $=\left\{B_{1},{\;B}_{2},\;\ldots .,{\;B}_{n}\;\right\}$ from PCAP, where B for Bytes
Compute  I, where I is Image
Decomposed I in SS1, where SS1=256 × 256
Apply FAST extractor & BRIEF descriptor on SS1Step 8:Generate texture features from the combination of FAST and BRIEFStep 9:Get Texture features as  T′Step 10:Combine T, T′ (Textual and texture features)Step 11:Apply SMOTE classing balancing on T,  T′
Compute BT, BT′ from T,  T′, where BT, BT′ are Balanced Texturaland Texture features 
BT=CNNT, to apply CNN technique of trained features
BT′=CNN T′, to apply CNN technique of texture featuresStep 12:Calculate Deep Features as DFStep 13:Apply Voting-based ensemble learning on DFStep 14:Result: Malware or BenignStep 15:Finish 

## 4. Results and Discussions

### 4.1. Dataset

We collected and prepared two datasets from the Canadian Institute for Cybersecurity (https://www.unb.ca/cic/datasets/index.html, accessed on 6 September 2021). The first dataset is the Investigation of the Android Malware (CIC-InvesAndMal2019) [33], which contains adware, botnet, premium SMS, ransomware, SMS, and scareware. They used real devices to install 5000 of the obtained samples (426 malware and 5065 benign). These samples were obtained from 42 distinct families of malware. Table 1 shows the first dataset. Table 2 shows the second dataset. The second dataset, CICMalDroid 2020 [34,35], contains 17,341 Android samples from VirusTotal, Contagio, AMD, and MalDozer. The samples were taken from 2017 to 2018. Further, the number of adware, banking, riskware, SMS, and benign are 1253, 2100, 2546, 3904, and 1795, respectively.

### 4.2. Results Analysis

Figure 7 depicts the epoch curves for malware classification, utilizing training and testing data in terms of accuracy, loss, precision, and recall. Parts a, b, c, and d depict the proposed model training and testing employing both datasets. The training curves for accuracy, loss, precision, and recall are represented by the colors blue, red, yellow, and green, respectively. Additionally, the same colors are utilized for testing curves employing the same performance metrics. In part a, precision begins at 60% and subsequently increases to 98% before becoming more or less constant for the model training using dataset 1. The recall curve begins at 70% and grows to 99% before becoming more or less constant with each epoch. The loss curve begins at 75% and gradually decreases until the 10th epoch. There is a tiny increase in the 18th epoch, but it is more or less consistent after that. It can be seen that the recall curve performs the best when compared to accuracy and precision. Part b depicts epoch curves for model testing with dataset 1. The accuracy, precision, and recall for test data range from 72% to 99%, respectively. While utilizing dataset 1, the largest and lowest testing loss is 75% and 8%, respectively. Part c depicts the model training for dataset 2. The three performance measures, namely accuracy, precision, and recall, perform similarly in the range of 70% to 99.4%. When compared to the other two metrics, the accuracy curve marginally rises on the 30th epoch. The model loss performance ranges from 70% to 5%. Part d depicts the model testing using dataset 2. The three performance measures begin at 73% and grow with each epoch. On the 15th and 30th epochs, performance suffers slightly. Following that, they exhibit more or less consistent behavior. The testing curve exhibits behavior ranging from 52% to 8%. The loss increases in the 5th, 15th, and 32nd epochs.

Figure 8 depicts the epoch curves for malware detection using the two datasets. Accuracy, precision, recall, and loss were used to demonstrate the efficacy of the proposed model. In part a, the accuracy, precision, and recall start at 78%, 28%, and 8%, respectively. On the 10th epoch, the accuracy and precision behave similarly, with a 99% accuracy. However, recall grows to 77% on the epoch and thereafter remains constant against each epoch. The loss curve starts at 82% and then drops to 8%. In part b, the accuracy and behavior range from 70% to 99.3%, while the recall ranges between 10% and 76%. Precision decreases slightly on the 4th epoch but increases again on the 10th. In comparison to the other two metrics, recall performed the worst. The testing loss begins at 85% and gradually increases to 90% in the 5th epoch. Following that, it drops with each epoch, with the lowest number being 15%. Part c depicts the training of the model for dataset 2. The three matrices exhibit behavior ranging from 48% to 99.2%. The recall begins at 48%, whereas the other two metrics begin at 75% and grow until the 12th epoch. Following that, the three curves remained relatively constant. The loss begins at 97% and gradually decreases from the 5th to the 38th epoch with a 3% loss. Recall performed the lowest when compared to the other two metrics. Similarly, part d depicts the model testing curves for dataset 2.

Table 3 shows the performance measures for malware detection using dataset 1. The precision, recall, and f1-score values of malware and benign classes were extracted. The ensemble model outperformed both classes, with a classification rate of 99%. The RF and DT performed better in both classes. The RF and DT metrics for malware class had 98%, 99%, 99%, 98%, 98%, and 99%, respectively. Similarly, the RF and DT had 99%, 98%, 99%, and 99%, 98%, 97%, respectively, for the benign class. However, GNB, SVM, and LR performed comparably for malware classes, while SVM and LR offered the worst results for benign classes. Overall, the ensemble model delivered the highest classification rates for malware classification using dataset 1 in terms of precision, recall, and f1-score. The malware detection using dataset 2 is shown in Table 4. It can be observed that the ensemble model delivered excellent classification rates for all three performance measures. Likewise, the RF and DT provided better classification results for both classifications, i.e., malware and benign, while the LR performed the worst.

To better understand the effectiveness of the proposed approach, we investigated performance measures such as precision, recall, and f1-score for each class. Figure 9 depicts the three malware classification performance measures using dataset 1. The colors blue, orange, and grey represent precision, recall, and f1-score, respectively. For all six malware families, the GNB, SVM, and LR performed the lowest. For instance, GNB and SVM had precision and f1-score values of 58% and 77%, respectively, while the recall rate for adware was 100%. Similarly, the precision, f1-score, and recall for the adware class using the LR method were 58%, 79%, and 100%, respectively. For all six malware families, the ensemble model outperformed, while, after the ensemble model, the DT and RF performed the second-best malware detection rates. Figure 10 depicts the malware classification performance measures using dataset 2. Precision provided the lowest performance for adware and banking, i.e., 60%, 62%, and 65% when using SVM, GNB, and LR, respectively, while recall performed the lowest values for banking, riskware, and SMS utilizing LR, GNB, and SVM, respectively. Overall, the ensemble model delivered the highest classification results for all malware families, whereas DT and RF were the second-best techniques.

The malware classification accuracy for each approach utilizing both datasets is shown in Table 5. When compared to the other approaches, the ensemble model had the highest classification and detection rates. The detection and classification accuracy of the ensemble model for dataset 1 was 98.44% and 99.16%, respectively. In dataset 2, these values were 97.76% and 98.91%, respectively. DT and RF provided the second-best performance. Overall, SVM was the worst performer, while GNB and LR were about average. Figure 11 depicts the confusion matrices for malware detection using both datasets. The parts from a to f represent dataset 1, while the parts from g to I represent dataset 2. The classification and misclassification rates are shown by the diagonal and off-diagonal values. In the first dataset, the GNB, SVM, DT, LR, RF, and ensemble classified malware and benign as (83.55%, 100%), (99.74%, 81.07%), (98.29%, 98.52%), (99.08%, 98.26%), and (99.08%, 99.19%), respectively. Similarly, using dataset 2, the malware and benign classification rates were (80.24%, 99.59%), (99.85%, 83.63%), (98.5%, 97.97%), (99.85%, 83.63%), (98.80%, 98.65%), and (98.95%, 98.78%), respectively. Figure 12 depicts the confusion matrices for the top three malware classification methods. Parts a to c depict dataset 1, while parts d to f depict dataset 2. Part a, adware, botnet, premium SMS, ransomware, scareware, and SMS malware detection rates were 98.56%, 98.07%, 97.82%, 97.2%, 98.71%, and 97.55%, respectively. In part c, using dataset 2, the detection rates for adware, banking, riskware, and SMS were 98.22%, 96.44%, 96.65%, and 97.83%, respectively. Using both datasets, we observed that the ensemble model provided the best detection rates.

### 4.3. Comparisons with Previous Methods

Table 6 compares precision, recall, f1-score, and accuracy to the current state of the approaches. The dataset was evaluated using four state-of-the-art algorithms, namely RNN, LSTM, DNN, and GRU, and the results were compared to the proposed approach. The RNN approaches had the lowest classification performance metrics, whereas the proposed approach had the highest. For instance, the proposed approach had precision, recall, f1-score, and accuracy rates of 99%, 99%, 99%, and 99.16%, respectively, while the RNN had 85%, 85%, 87%, and 85.34%, respectively. For instance, the precision, recall, f1-score, and accuracy of the proposed approach were 99%, 99%, 99%, and 99.16%, respectively, whereas the RNN had 85%, 85%, 87%, and 85.34%, respectively. The DNN was the second-best strategy for producing better classification results. 

Table 7 shows the performance comparison with recently published works. These studies mostly made use of network traffic to classify Android malware. The botnet malware was classified using HTTP traffic [17]. It analyzes network traffic to build malware groups. This approach can detect new clustered malware with 98.66% precision. Droid Classifier automatically develops numerous models over labeled malicious apps [20]. Each model uses network-traffic IDs. Automatic threshold parameters are intended to properly describe various malware features with a 94.66% accuracy. In the [36] study, URLs were used to locate malicious programs. For malware analysis, multi-view neural networks provide depth and breadth. It develops and distributes soft attention-weighting elements for data with an accuracy of 95.74%. The static and dynamic methods are used to monitor and examine infected Android apps. Overall, the Android malware detection rate is 98.86%. URLs are used to detect Android malware [37]. Malware detection models with deep features are built with multi-view neural networks. The feature weights are spread across inputs and provide an accuracy of 98%. In the study [38], seven supervised learning strategies were investigated and compared to determine how the method works. CNN classifiers include two-layer CNN, four-layer CNN, VGG16, LR, SVM, and K-NN. In experiments, the SVM classifier reached a 94% accuracy. The proposed ensemble technique outperformed with a malware detection accuracy of 99.16%.

The presented method was carefully compared with other approaches that have already been used with the same datasets, as shown in Table 8. Malware can be classified using texture-based, text-based, or a combined effect of both. Alani et al. [40] presented AdStop, a machine learning-based technique for detecting vulnerabilities in network data. The proposed technique classified malware with a 98.2% accuracy using AAGM2017 word embeddings and a deep neural network. The framework suggested by Saket et al. [41] uses hierarchical and Latent Dirichlet Allocation techniques to extract clusters. They classified malware using the CNN model, which has a precision of 98.3%, without using any specialized features. The [41,42,43,44] classified malware utilized texture characteristics using CNN and Temporal Convolutional Network models. The suggested models classify malware using images without utilizing descriptors to identify their unique features. The studies [34,40,45] used deep neural networks, gradient boosting, and ensemble learning to classify malware based on text-based features. We proposed a technique for classifying malware that integrated text-based and texture-based features from both datasets. Our proposed method surpassed state-of-the-art approaches with a classification result of 99%.

### 4.4. Model Interpretation and Validation Using Explainable AI

To interpret and validate the proposed approach, an explainable AI approach can be used. We used the Local Interpretable Model-agnostic Explanation (LIME) and Shapley Additive exPlanations (SHAP) libraries to describe the impact of each feature on the accuracy of the model [46]. SHAP values indicate how much evidence a feature provides to the output of a model. Figure 13 shows the SHAP values of each feature shift model output from our initial expectation to the final model prediction. The features are ordered according to their SHAP values, with the smallest values placed together at the bottom of the maximum display being exceeded. The red and blue colors represent the contribution of malware and benign features, respectively. We had a total of 32 features, with the influence of the top nine features highlighted. The F27 with a SHAP value of 209 was the most important feature that contributed to the malware class, while F22 contributed to the benign class with a SHAP value of 90. It demonstrated that F27 and F22 features were the most important contributors to malware and benign classes. Figure 14 depicts the relative impact of features on obtaining an outcome of 1 (benign) from an aggregate of observations with a threshold value of 0 (malware). The threshold of 1.01 splits the impact of features into two categories, namely malware and benign. The colors red and blue represent the contribution of features to malware and benign. We discovered that the F27 feature contributed significantly to malware, while the F22 feature contributed significantly to the benign class. This allows us to readily mine those features that contribute significantly to the model’s performance.

We utilized absolute values to avoid having positive and negative SHAP values cancel each other out. Figure 15 depicts the primary effects and interaction effects of the first ten features. For example, we can observe that the average main effect for F1, F2, F3, F8, and F9 is high. This indicates that these features are more likely to have significant positive or negative main effects. In other words, these characteristics have a big influence on the model’s results. Similarly, the F1, F2, F3, F8, and F9 interaction effects are also large. Figure 16 depicts the interaction of features concerning the model’s output. The SHAP values for the primary impacts are presented on the diagonals, while the interaction impacts are shown on the off-diagonals. This can provide insight into absolute mean values by highlighting main and interaction effects. For instance, the high SHAP values correspond to high absolute mean values. Additional information can be gained by examining the relationships depicted in the interaction values. For instance, we can observe that the F1, F2, F3, F8, and F9 all have favorable major effects.

## 5. Conclusions

Android has quickly become the most popular mobile operating system due to its versatility and user-friendliness. The majority of persistent malicious attackers make it their main target as well. This calls for the immediate implementation of a robust malware detection system. This study proposed an Android malware detection method that makes use of BERT-based transfer learning and graphical malware features. The PCAP file contains extensive information about Android network communication. This folder contains files that are used to record HTTP and TCP network traffic. Using the pre-trained BERT model, the trained features are extracted from a large number of textual features. To extract visual features, the byte stream of a PCAP file is converted into a standard 256 × 256 image. Furthermore, the FAST extractor and BRIEF descriptor are used to extract and label key features efficiently. The deep feature representations are then collected using the CNN network. We encountered a class imbalance issue because of the mix of textual and texture features. The SMOTE approach is utilized to balance the features, which are then input into the ensemble model for malware detection and classification. The proposed work provided malware detection and classification rates of 98.44% and 99.16%, respectively, using dataset 1. The results were similar using dataset 2. The malware detection and classification rates were 97.76% and 98.91%, respectively. Using an explainable AI methodology, the proposed method was interpreted and validated. 

In the future, GloVe and Fast-text trained models would be used to mine the trained features. Furthermore, advanced deep learning models such as multi-head and reinforcement learning can be utilized to evaluate malware detection performance.

## Figures and Tables

**Figure 1 sensors-22-06766-f001:**
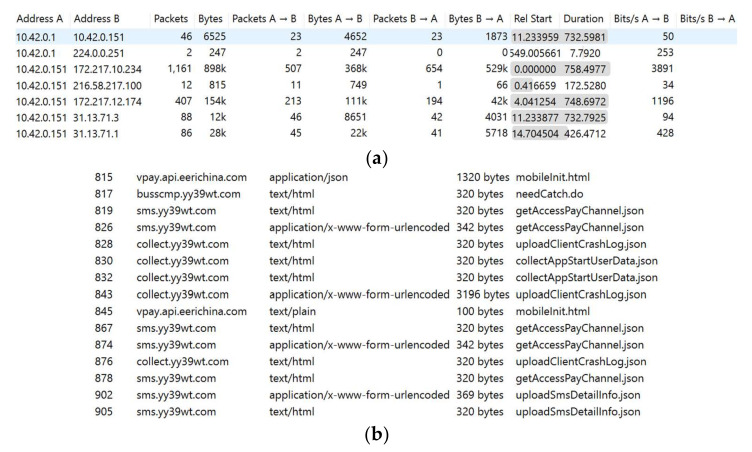
Malicious network patterns using adware and SMS malware: (**a**) malicious unresolved IP using adware; (**b**) malicious scripts using SMS malware.

**Figure 2 sensors-22-06766-f002:**
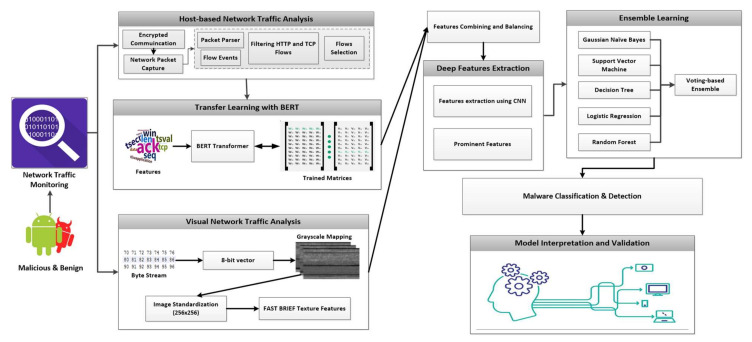
Explainable malware detection system using transformer-based transfer learning and visual features.

**Figure 3 sensors-22-06766-f003:**
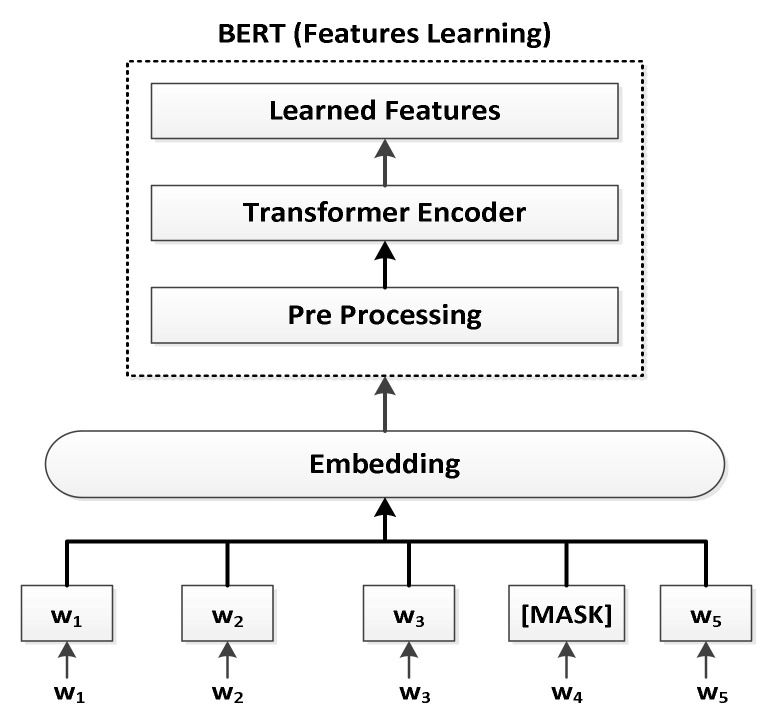
Feature mapping with BERT.

**Figure 4 sensors-22-06766-f004:**
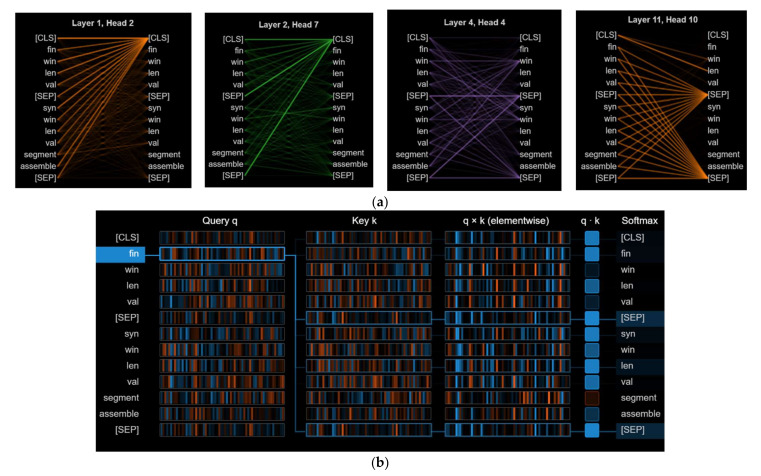
Visualization attention for network traffic data using the BERT pre-trained model: (**a**) head view visualization attention with different layers using Tensor2Tensor; (**b**) neuron view visualization between Query q and Key k.

**Figure 5 sensors-22-06766-f005:**
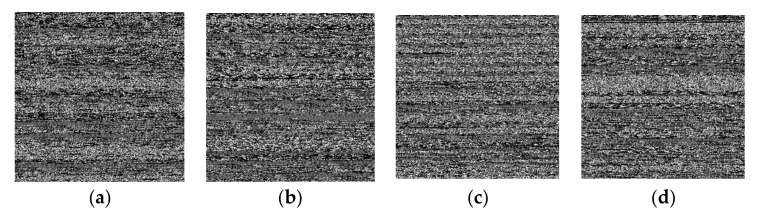
Malware images extracted from network traffic with a size 256 × 256: (**a**) Botnet; (**b**) Premium SMS; (**c**) Ransomware; (**d**) Scareware.

**Figure 6 sensors-22-06766-f006:**
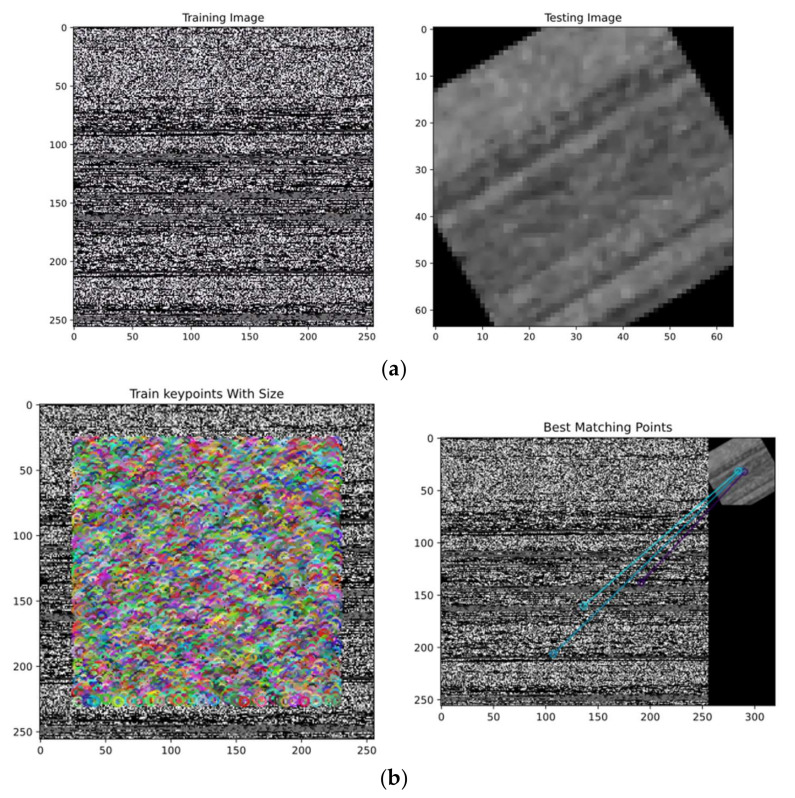
Tracking of features between train and test malware images: (**a**) train and test images (scale and rotation invariant); (**b**) train key points with the best matching points.

**Figure 7 sensors-22-06766-f007:**
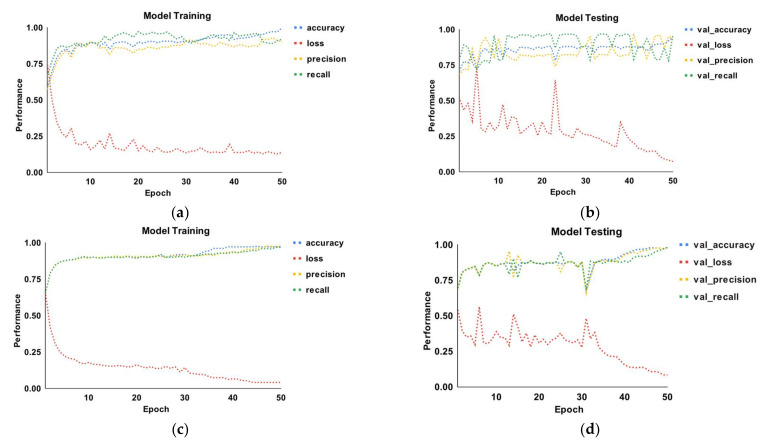
Dynamic epoch curves (accuracy, precision, recall, loss) for malware classification using training and testing data: (**a**) dataset 1 (model training); (**b**) dataset 1 (model testing); (**c**) dataset 2 (model training); (**d**) dataset 2 (model testing).

**Figure 8 sensors-22-06766-f008:**
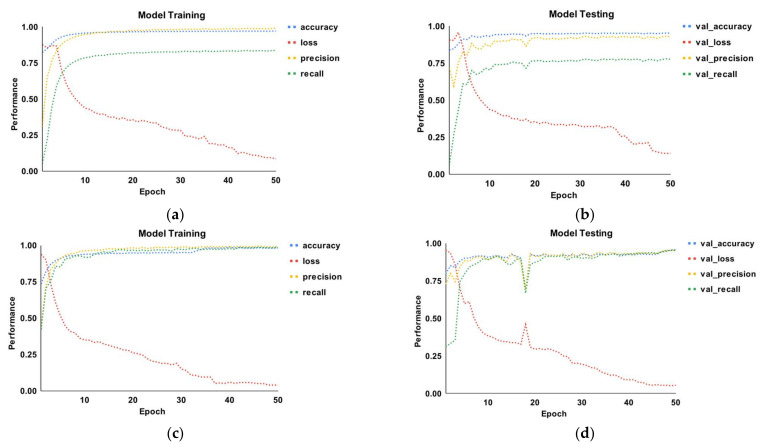
Dynamic epoch curves (accuracy, precision, recall, loss) for malware detection using training and testing data: (**a**) dataset 1 (training); (**b**) dataset 1 (testing); (**c**) dataset 2 (training); (**d**) dataset 2 (testing).

**Figure 9 sensors-22-06766-f009:**
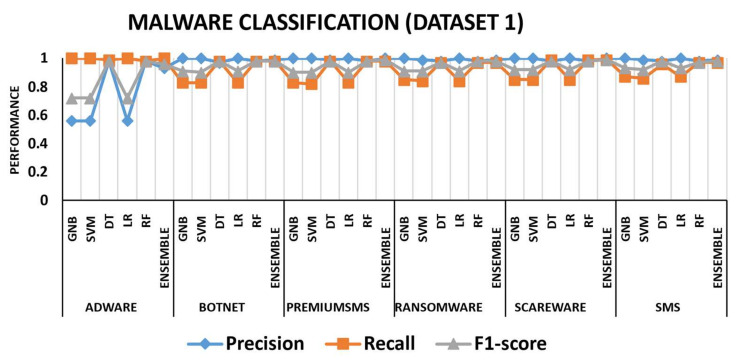
Performance measure (precision, recall, f1-score) comparisons for malware detection dataset 1.

**Figure 10 sensors-22-06766-f010:**
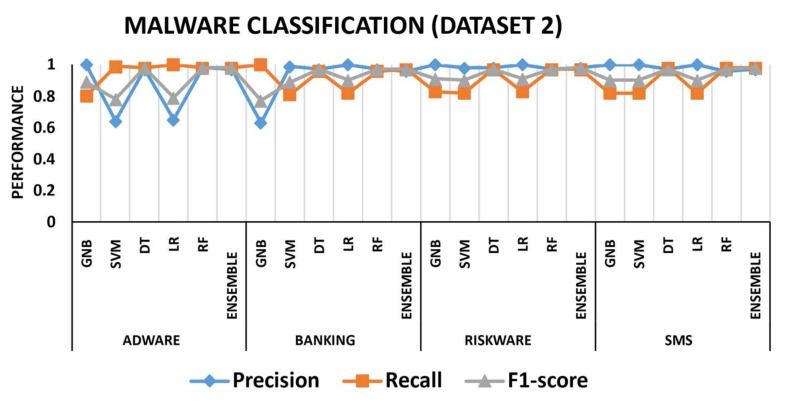
Performance measure (precision, recall, f1-score) comparisons for malware detection dataset 2.

**Figure 11 sensors-22-06766-f011:**
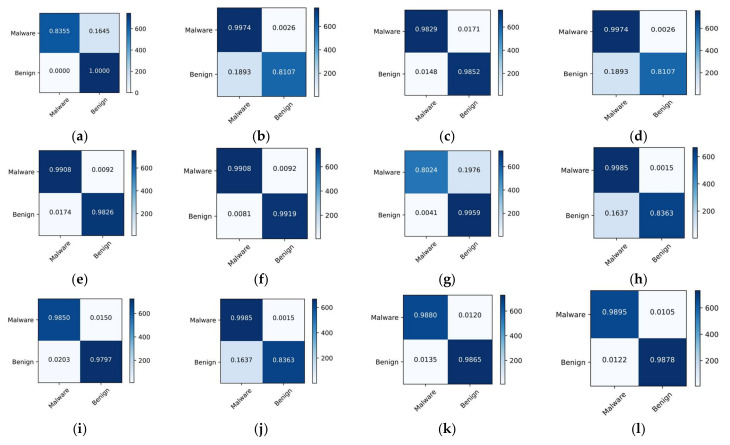
Confusion matrices for malware detection using both datasets: (**a**) GNB; (**b**) SVM; (**c**) DT; (**d**) LR; (**e**) RF; (**f**) Ensemble; (**g**) GNB; (**h**) SVM; (**i**) DT; (**j**) LR; (**k**) RF; (**l**) Ensemble.

**Figure 12 sensors-22-06766-f012:**
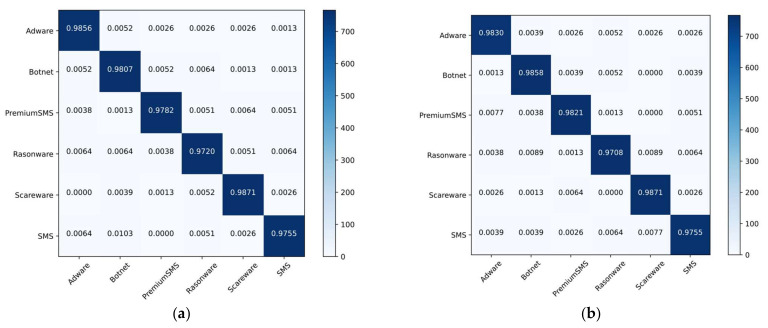
Confusion matrices for malware classification using both datasets for the top three algorithms: (**a**) DT; (**b**) RF; (**c**) RF; (**d**) DT; (**e**) RF; (**f**) Ensemble.

**Figure 13 sensors-22-06766-f013:**
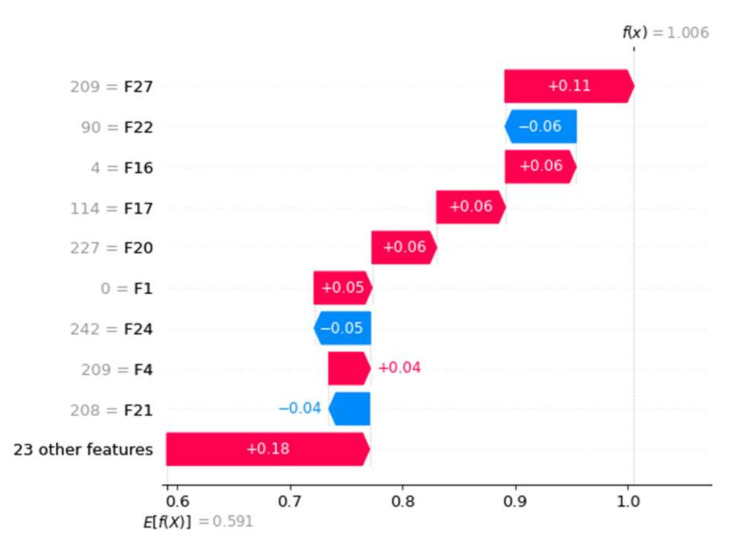
Waterfall plot shows how SHAP values of each feature affect model output compared to the data distribution.

**Figure 14 sensors-22-06766-f014:**
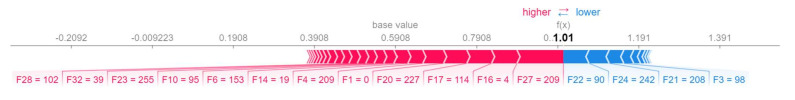
Contribution of features to a specific class depending on a threshold level.

**Figure 15 sensors-22-06766-f015:**
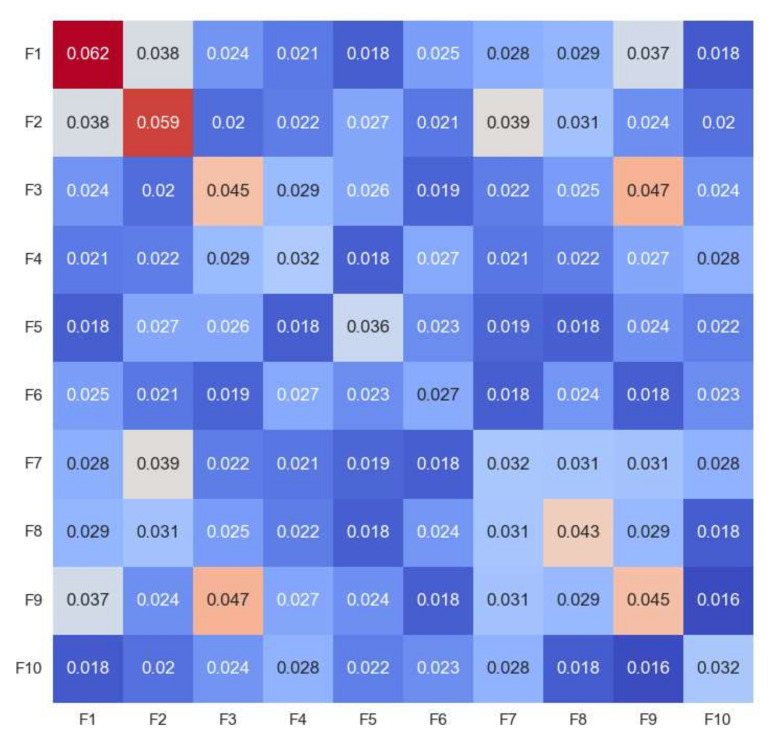
Absolute mean of main and interaction effects for first 10 features.

**Figure 16 sensors-22-06766-f016:**
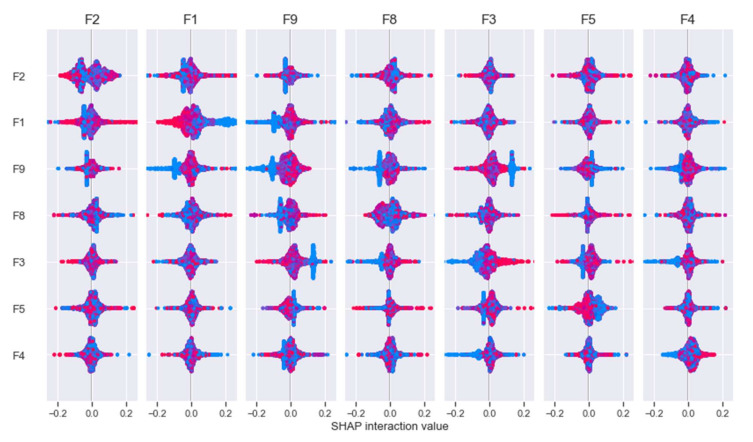
Interaction of features for the model output.

**Table 1 sensors-22-06766-t001:** CIC-InvesAndMal2019 dataset (dataset 1).

Apps	Families	Description
Malware	Adware	There are ads concealed within malware-infected programs
	Botnet	Performs DDoS attacks, steals data, and provides access to the device
	Premium SMS	SMS fraud exploits mobile premium service billing
	Ransomware	Restricts computer files until the ransom is paid
	Scareware	Scares users into visiting fake sites or installing malware
	SMS	Conducts attacks through the SMS alert
Benign	Benign	Clean apps (not malicious)

**Table 2 sensors-22-06766-t002:** CICMalDroid 2020 dataset (dataset 2).

Apps	Families	No. of Apps	Description
Malware	Adware	1253	There are ads concealed within malware-infected programs
	Banking	2100	Authenticates their internet banking services
	Riskware	2546	Can be any legitimate app that, if exploited, can bring harm
	SMS	3904	Conducts cyberattacks through an SMS alert
Benign	Benign	1795	Clean apps (not malicious)

**Table 3 sensors-22-06766-t003:** Performance measures for malware detection using dataset 1.

	Methods	Precision (%)	Recall (%)	F1-Score (%)
Malware	GNB	100	84	91
	SVM	84	100	91
	DT	98	98	99
	LR	84	100	91
	RF	98	99	99
	**Ensemble**	99	99	99
Benign	GNB	86	100	92
	SVM	100	81	89
	DT	99	98	97
	LR	100	81	89
	RF	99	98	99
	**Ensemble**	99	99	99

**Table 4 sensors-22-06766-t004:** Performance measures for malware detection using dataset 2.

	Methods	Precision (%)	Recall (%)	F1-Score (%)
Malware	GNB	99	80	89
	SVM	85	100	92
	DT	97	97	98
	LR	85	100	92
	RF	97	98	98
	**Ensemble**	97	99	98
Benign	GNB	85	100	92
	SVM	100	84	91
	DT	98	97	98
	LR	100	84	91
	RF	98	98	98
	**Ensemble**	99	98	98

**Table 5 sensors-22-06766-t005:** Malware detection and classification accuracy using both datasets.

	Methods	Detection (%)	Classification (%)
**Dataset 1**	GNB	87.21	91.55
	SVM	87.34	90.12
	DT	97.94	98.36
	LR	88.1	90.08
	RF	98.2	98.54
	**Ensemble**	98.44	99.16
**Dataset 2**	GNB	86.92	89.99
	SVM	87.1	91.80
	DT	97.2	98.34
	LR	87.58	91.76
	RF	97.34	98.69
	**Ensemble**	97.76	98.91

**Table 6 sensors-22-06766-t006:** Comparisons with state-of-the-art methods.

Method	Precision (%)	Recall (%)	F1-Score (%)	Accuracy (%)
RNN	85	85	87	85.34
LSTM	82	80	81	80.02
DNN	88	87	85	86.16
GRU	80	78	80	78.62
Our Approach	99	99	99	99.16

**Table 7 sensors-22-06766-t007:** Comparisons with published works.

Work	Year	Methods	Accuracy (%)
**Aresu et al.** [17]	2015	Signature-based clustering	96.66
**Li et al.** [39]	2016	Droid classifier	94.66
**Shanshan et al.** [36]	2018	Skip-gram with neural network	95.74
**Shanshan et al.** [10]	2019	C4.5 decision tree	97.89
**Shyong et al.** [4]	2020	Random forest	98.86
**Shanshan et al.** [37]	2020	Multi-view neural network	98.81
**Rania et al.** [38]	2021	Machine learning with CNN	94
**Our approach**	2022	BERT-based transfer learning and visual representation	99.16

**Table 8 sensors-22-06766-t008:** Performance comparison with state-of-the-art methods using the same datasets.

Work	Dataset	Strategy	Method	Accuracy (%)
**Alani et al.** [40]	AAGM2017	Text-based	Deep neural network	98.02
**Saket et al.** [41]	AAGM2017	Texture-based	Convolutional neural network	98.3
**Raden et al.** [45]	MalDroid 2020	Text-based	Gradient boosting	96.35
**Fawareh et al.** [42]	MalDroid 2020	Texture-based	Convolutional neural network	96.4
**Wenhui et al.** [43]	MalDroid 2020	Texture-based	Temporal convolutional network	95.44
**Samaneh et al.** [34]	MalDroid 2020	Text-based	Ensemble learning	97.84
**Tao et al.** [44]	MalDroid 2020	Texture-based	Convolutional neural network	98.6
**Our proposed**	AAGM2017 & MalDroid 2020	Text and Texture	BERT-based transfer learning and visual representation	99

## Data Availability

The data that support the findings of this study are openly available in Canadian Institute for Cybersecurity-CIC-InvestAndMal2019 and CICMalDroid2020 at https://www.unb.ca/cic/datasets/invesandmal2019.html (accessed on 6 September 2021), and https://www.unb.ca/cic/datasets/maldroid-2020.html (accessed on 6 September 2021), respectively.

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
