# Peer review of "Explainable Malware Detection System Using Transformers-Based Transfer Learning and Multi-Model Visual Representation"

_sensors, 2022, doi:10.3390/s22186766_

Round 1
Reviewer 1 Report
This study suggests that a malware detection system should employ visual characteristics and transfer learning. Malware is identifiable by its textual and visual characteristics. Initially, a BERT model extracts features from the text. The malware-to-image program transforms network bytes into images. The terms FAST and BRIEF are useful for locating and highlighting significant features. CNN extracts deep features by combining trained features with texture features using SMOTE. An ensemble model, which is used to classify and identify malware, is fed with balanced features. Putting explainable AI to use in the malware classification space is an interesting prospect.
• In introduction section: “Android is the most popular smartphone operating system in the world, with over 80% of the market share”. Needs to add url/reference of the study.
• The main theme is about malicious network traffic using text and visual features of PCAPs. It needs to filter important information. The data preprocessing section needs to be strengthened.
• There is a typo error in figures 9 and 10. These figures are about malware classification (detection can be replaced by classification).
• The comparisons demonstrate cutting-edge methods and published works. I would recommend to do experiment/s for comparing the proposed work to previously published works using the same datasets.
• The conclusion contains insufficient granularity of information. I suggest to suggest to add the statistical evidence to prove the claim.
Author Response
Response to Reviewer 1 Comments
This study suggests that a malware detection system should employ visual characteristics and transfer learning. Malware is identifiable by its textual and visual characteristics. Initially, a BERT model extracts features from the text. The malware-to-image program transforms network bytes into images. The terms FAST and BRIEF are useful for locating and highlighting significant features. CNN extracts deep features by combining trained features with texture features using SMOTE. An ensemble model, which is used to classify and identify malware, is fed with balanced features. Putting explainable AI to use in the malware classification space is an interesting prospect.
Point 1: In introduction section: “Android is the most popular smartphone operating system in the world, with over 80% of the market share”. Needs to add url/reference of the study.
Response 1: Thanks. The reference [1] is added in the revised manuscript.
- Liu, Pei, Li Li, Yanjie Zhao, Xiaoyu Sun, and John Grundy. "Androzooopen: Collecting large-scale open source android apps for the research community." In Proceedings of the 17th International Conference on Mining Software Repositories, pp. 548-552. 2020.
Point 2: The main theme is about malicious network traffic using text and visual features of PCAPs. It needs to filter important information. The data preprocessing section needs to be strengthened.
Response 2: Thanks. The data preprocessing step is dicussed in detailed and add the following information. The paper is revised accordingly.
PCAP files are the primary records that are created during network data transmission. This document contains network traffic used to evaluate the malicious node communication process. They also aid in network traffic planning and activity sensing. HTTP traces include source, destination, port, host, source info, bytes, packet length, frame length, and TTL. GET, POST, and URLs like "www.google.com" are in the source info. TCP flows include transmitted and received data and overall session counts throughout conversations. Valuable information can be filtered to preserve semantics. To prevent redundant data, eliminate attributes from input sequences that are consecutively similar. Short patterns are removed because they may not provide enough data to recognize network activity. Unifying sequence is crucial for detecting attacks because distinct pattern dimensions mislead neural network algorithms. To adjust the dimension, this method employs a predefined sequence length L. Sequences longer than L retain their first L names, whereas those shortened than L are unified by zero-padding.
Point 3: There is a typo error in figures 9 and 10. These figures are about malware classification (detection can be replaced by classification).
Response 3: Thanks. The typo error in Figures 9 and 10 are fixed. Moreover, some other typo errors are also fixed regarding classification and detection keywords. The paper is revised accordingly.
Point 4: The comparisons demonstrate cutting-edge methods and published works. I would recommend to do experiment/s for comparing the proposed work to previously published works using the same datasets.
Response 4: Thanks. The proposed work is compared with the existing works using the same datasets. The following information is added in the revised paper.
The presented method is carefully compared with other approaches that have already been used with the same datasets as shown in Table 8. Malware can be classified using texture-based, text-based, or a combined effect of both. Alani et al. [2] presented AdStop, a machine learning-based technique for detecting vulnerabilities in network data. The proposed technique classified malware with 98.2% accuracy using AAGM2017 word embeddings and a deep neural network. The framework suggested by Saket et al. [3] uses hierarchical and Latent Dirichlet Allocation techniques to extract clusters. They classified malware using the CNN model, which has a precision of 98.3%, without using any specialized features. The [3-6] classified malware utilizing texture characteristics using CNN and Temporal Convolutional Network models. The suggested models classify malware using images without utilizing descriptors to identify their unique features. The [2, 7, 8] used deep neural networks, gradient boosting, and ensemble learning to classify malware based on text-based features. We proposed a technique for classifying malware that integrated text-based and texture-based- features from both datasets. Our proposed method surpasses state-of-the-art approaches with a classification result of 99%.
Table 8: Performance comparison with state-of-the-art methods using same datasets
|
Work |
Dataset |
Strategy |
Method |
Accuracy (%) |
|
Alani et al. [2] |
AAGM2017 |
Text-based |
Deep Neural Network |
98.02 |
|
Saket et al. [3] |
AAGM2017 |
Texture-based |
Convolutional Neural Network |
98.3 |
|
Raden et al. [4] |
MalDroid 2020 |
Text-based |
Gradient Bossting |
96.35 |
|
Fawareh et al. [5] |
MalDroid 2020 |
Texture-based |
Convolutional Neural Network |
96.4 |
|
Wenhui et al. [6] |
MalDroid 2020 |
Texture-based |
Temporal Convolutional Network |
95.44 |
|
Samaneh et al. [7] |
MalDroid 2020 |
Text-based |
Ensemble Learning |
97.84 |
|
Tao et al. [8] |
MalDroid 2020 |
Texture-based |
Convolutional Neural Network |
98.6 |
|
Our proposed |
AAGM2017 & MalDroid 2020 |
Text and Texture |
BERT-based transfer leaning and visual representation |
99 |
- Alani, M.M. and A.I. Awad, AdStop: Efficient flow-based mobile adware detection using machine learning. Computers & Security, 2022. 117: p. 102718.
- Acharya, S., U. Rawat, and R. Bhatnagar, A Low Computational Cost Method for Mobile Malware Detection Using Transfer Learning and Familial Classification Using Topic Modelling. Applied Computational Intelligence and Soft Computing, 2022. 2022.
- Al-Fawa'reh, M., et al. Malware detection by eating a whole APK. in 2020 15th International Conference for Internet Technology and Secured Transactions (ICITST). 2020. IEEE.
- Zhang, W., et al., Android malware detection using tcn with bytecode image. Symmetry, 2021. 13(7): p. 1107.
- Peng, T., et al., A Lightweight Multi-Source Fast Android Malware Detection Model. Applied Sciences, 2022. 12(11): p. 5394.
- Hadiprakoso, R.B., H. Kabetta, and I.K.S. Buana. Hybrid-based malware analysis for effective and efficiency android malware detection. in 2020 International Conference on Informatics, Multimedia, Cyber and Information System (ICIMCIS). 2020. IEEE.
- Mahdavifar, S., et al. Dynamic android malware category classification using semi-supervised deep learning. in 2020 IEEE Intl Conf on Dependable, Autonomic and Secure Computing, Intl Conf on Pervasive Intelligence and Computing, Intl Conf on Cloud and Big Data Computing, Intl Conf on Cyber Science and Technology Congress (DASC/PiCom/CBDCom/CyberSciTech). 2020. IEEE.
Point 5: The conclusion contains insufficient granularity of information. I suggest to suggest to add the statistical evidence to prove the claim.
Response 5: The conclusion section is update with the required statistical information. The paper is revised accordingly.
Reviewer 2 Report
Title: Explainable malware detection system fusing transformers-based transfer learning and multi-model visual representation This study proposes a malware detection system using transfer learning and visual features. The technique uses textual and visual features to detect malware. First, a pre-trained BERT model extracts trained textual features. Second, the malware-to-image conversion algorithm transforms network bytes into images. The FAST extractor and BRIEF descriptor efficiently extract and mark important features. CNN network is then used to mine the deep features after the trained and texture features have been combined and balanced using the SMOTE. Balanced features are fed into an ensemble model for malware classification and detection. CICMalDroid 2020 and CIC-InvesAndMal2019 are used to analyze the proposed method. An AI experiment is conducted to explain and validate the proposed methodology. Minor revisions are required before accepting for publication. This paper is well written and clearly addresses the problem. The problem is important and the outcomes are promising. The explainable AI part enhances the interpretation of the proposed work. The BERT model is used to extract textual information from network traffic. What kind of BERT is being used? write the number of hidden layers, attention heads, and transform block encoded layers. How ensemble selects the best classification results? Explain the working of voting ensemble in section 3.5. As mentioned, the Algorithm 1 describes the complete proposed methodology. It needs more explanation. The paper should be checked to make sure it is formatted correctly and that the figures are easy to read.Author Response
Response to Reviewer 2 Comments
Title: Explainable malware detection system fusing transformers-based transfer learning and multi-model visual representation This study proposes a malware detection system using transfer learning and visual features. The technique uses textual and visual features to detect malware. First, a pre-trained BERT model extracts trained textual features. Second, the malware-to-image conversion algorithm transforms network bytes into images. The FAST extractor and BRIEF descriptor efficiently extract and mark important features. CNN network is then used to mine the deep features after the trained and texture features have been combined and balanced using the SMOTE. Balanced features are fed into an ensemble model for malware classification and detection. CICMalDroid 2020 and CIC-InvesAndMal2019 are used to analyze the proposed method. An AI experiment is conducted to explain and validate the proposed methodology. Minor revisions are required before accepting for publication.
Point 1: This paper is well written and clearly addresses the problem.
Response 1: Thanks for the encouraging comment.
Point 2: The problem is important and the outcomes are promising.
Response 2: Thanks for the encouraging comment.
Point 3: The explainable AI part enhances the interpretation of the proposed work.
Response 3: Thanks for the encouraging comment.
Point 4: The BERT model is used to extract textual information from network traffic. What kind of BERT is being used? write the number of hidden layers, attention heads, and transform block encoded layers.
Response 4: We employed a BERT-based model [1] for word embedding and transfer learning from network traffic. It employs 12 layers of transformer blocks, has a hidden size of 768, 12 self-attention heads, and approximately 110M trainable parameters. The information is added in the revised paper.
- Gao, Z., et al., Target-dependent sentiment classification with BERT. Ieee Access, 2019. 7: p. 154290-154299.
Point 5: How ensemble selects the best classification results? Explain the working of voting ensemble in section 3.5.
Response 5: In soft voting each independent classifier provides a statistically significant indication that a given data point belongs to a particular class label. This enables more progressive and decentralized decision-making. The predictions are summed after being weighted in accordance with the importance of the classification model. The vote is then given to the target class label with the highest sum of normalized probabilities [2]. The section 3.5 is revised accordingly.
- Ahmed, U., J.C.-W. Lin, and G. Srivastava, Mitigating adversarial evasion attacks of ransomware using ensemble learning. Computers and Electrical Engineering, 2022. 100: p. 107903.
Point 6: As mentioned, the Algorithm 1 describes the complete proposed methodology. It needs more explanation.
Response 6: It describes the overall procedure for the proposed study. The network traffic is provided as input in the form of PCAPs, and the malware classification is delivered as output. The PCAP file is filtered for the required TCP and HTTP information. The BERT-base model is intended to extract train features from the combination of TCP and HTTP. PCAP bytes are mined and converted to images to extract texture features using FAST and BRIEF. Textural and texture features are combined and fed into the soft-based voting ensemble model for effective malware detection and classification.
Point 7: The paper should be checked to make sure it is formatted correctly and that the figures are easy to read.
Response 7: Thanks for the comment. The revised paper is checked for formatting, and readability of figures.
Round 2
Reviewer 1 Report
My comments are addressed however there are some minor changes needed.
About preprocessing step, how the proposed work set the features in order to process by the BERT model efficiently?
Author Response
Response to Reviewer 1 Comments
Point 1: My comments are addressed however there are some minor changes needed. About preprocessing step, how the proposed work set the features in order to process by the BERT model efficiently?
Response 1: The BERT-based mapping of the features is depicted in Figure 3. A sequence of embedded network features (w1, w2, etc.) are processed by the BERT-based neural network. Each of the resultant H-dimensional vectors corresponds to an input feature with the same index. Before feeding each feature sequence into BERT, 15% of the features are replaced with [MASK] tokens. The relevance of the non-masked features is used by the model to forecast the current value of the masked features. We used the following parameters for BERT output feature prediction.
- A classification layer is added on top of the encoder output.
- By multiplying the output vectors by the embedding matrix, the lexical dimension is made from the output vectors.
- The probability of each feature in the vocabulary is calculated with the help of the softmax method.
Moreover, we used the following pre-trained repositories to preprocess the network traffic.
bert_preprocess = hub.KerasLayer("https://tfhub.dev/tensorflow/bert_en_uncased_preprocess/3")
bert_encoder = hub.KerasLayer("https://tfhub.dev/tensorflow/bert_en_uncased_L-12_H-768_A-12/4")
